# Identification and pathogen detection of a *Neocypholaelaps* species (Acari: Mesostigmata: Ameroseiidae) from beehives in the Republic of Korea

Thi-Thu Nguyen[1,2☯], Mi-Sun Yoo[1☯], Jong-Ho Lee[3], A-Tai Truong[1,4], So-Youn Youn[1], Se-Ji Lee[1], Soon-Seek Yoon[1], Yun Sang Cho[1]*

1 Laboratory of Parasitic and Honeybee Diseases, Bacterial Disease Division, Department of Animal and Plant Health Research, Animal and Plant Quarantine Agency, Gimcheon, Republic of Korea, 2 Institute of Biotechnology, Vietnam Academy of Science & Technology, Ha Noi, Viet Nam, 3 Plant Pest Control Division, Department of Plant Quarantine, Animal and Plant Quarantine Agency, Gimcheon, Republic of Korea, 4 Faculty of Biotechnology, Thai Nguyen University of Sciences, Thai Nguyen, Viet Nam

☯ These authors contributed equally to this work.
* choys@korea.kr

**Data Availability Statement:** The amplified 18S and 28S ribosomal DNA gene sequences of

## Abstract

In this study, we identified a new strain of the genus *Neocypholaelaps* from the beehives of *Apis mellifera* colonies in the Republic of Korea (ROK). The *Neocypholaelap* sp. KOR23 mites were collected from the hives of honeybee apiaries in Wonju, Gangwon-do, in May 2023. Morphological and molecular analyses based on 18S and 28S rRNA gene regions conclusively identified that these mites belong to the genus *Neocypholaelaps*, closely resembling *Neocypholaelaps* sp. APGD-2010 that was first isolated from the United States. The presence of 9 of 25 honeybee pathogens in these mite samples suggests that *Neocypholaelaps* sp. KOR23 mite may act as an intermediate vector and carrier of honeybee diseases. The identification of various honeybee pathogens within this mite highlights their significance in disease transmission among honeybee colonies. This comprehensive study provides valuable insights into the taxonomy and implications of these mites for bee health management and pathogen dissemination.

## Introduction

During the current honeybee disease outbreak, parasites played a significant role in contributing to the sudden collapse of the colony. Several species of bee mites can coexist with honeybee (*Apis mellifera*) colonies for extended periods. In *A. mellifera*, some mite species may be not exhibit parasitic behavior (e.g. *Forcellinia faini*, *Melichares dentriticus*, *Afrocypholaelaps africana* mites) [1–4] or transmit diseases to bees (e.g. *Varroa* and *Tropilaelaps* mites) [5–7], a considerable number of mite species can cause stress to bees (e.g. *Neocypholaelaps indica* and *Carpoglyphus lactis* mites) [8, 9], resulting in reduced flight capacity and foraging efficiency. Additionally, certain parasitic mite species infest bees, causing injury or mortality in their

*Neocypholaelaps* sp. KOR23 obtained in this study were deposited in the National Center for Biotechnology Information (NCBI) GenBank database under accession numbers OR576776.1 and OR576780.1, respectively.

**Funding:** This work was supported by Animal and Plant Quarantine Agency (Grant No. N-1543081-2021-25-03).

**Competing interests:** No authors have competing interests

hosts and facilitating the spread of harmful diseases. Obligate honeybee parasites like *Varroa* (Varroidae, Mesostigmata), *Euvarroa* (Varroidae, Mesostigmata), *Tropilaelaps* (Laelapidae, Mesostigmata), and *Acarapis* (Tarsonemidae, Prostigmata inflict direct harm and facilitate disease spread [10–12]. These mites have co-evolved with their hosts, developing an intimate dependence on honeybees for survival. While *Varroa* mites primarily parasitize bees in their brood cells [6, 13], *Euvarroa* and *Tropilaelaps* demonstrate host specificity towards Dwarf honeybees (*Apis florea)* and giant honeybees (*Apis dorsata)*, respectively [7, 14–17]. The most pathogenic species within the genus *Acarapis* primarily parasitize *A. mellifera* but have also been found in Asian honeybee species [18–22]. Interestingly, the genus *Tyrophagus* (Acaridae: Sarcoptiformes) adds to the list of potentially detrimental mites, displaying parasitic behavior detrimental to both honeybees and bumblebees [23, 24].

The genus *Neocypholaelaps*, encompassing 22 species primarily concentrated in tropical regions, continues to pique the interest of researchers due to its diverse interactions with honeybees [25–40]. These *Neocypholaelaps* mites, belonging to the family Ameroseiidae, exhibit a relationship with their bee hosts, often associating with them both externally on their bodies and internally within their nests and hives [31]. Meanwhile, several *Neocypholaelaps* species, including *N. favus* Ishkawa, 1968, *N. apicola* Delfinado-Baker & Baker, 1983, *N. geonomae* Moraes & Narita, 2010, and *N. indica* Evans, 1963 have been identified in honeybee colonies across various geographical locations, spanning Europe, India, China, and even regions beyond the tropics, highlighting their adaptability [27, 32–35]. *N. apicola* Delfinado-Baker & Baker, 1983 was reported on the bodies of bees and within honeybee colonies in multiple European countries such as Greece [35], Denmark [36], Belgium [33], Slovakia [37], and Hungary [38], demonstrating its wider distribution. Another species, *N. indica* Evans, 1963, has been reported in both *A. mellifera* and *A. cerana* in Fuzhou, China, and southern Karnataka, India [39]. *N. stridulans* Evans, 1955 interacts with mason bees and coconut flower clusters in India [40]. In the Republic of Korea (ROK), the detection of *Neocypholaelaps* mites in honeybee colonies has also been reported in Gangwon and Gyeonggi provinces; however, species classification remains unclear.

Based on morphological characteristics, it is possible to determine the family to which a species belongs; however, accurate species-level identification can be challenging. Molecular methods, particularly DNA sequence-based identification, have proven to be the most reliable approaches for precise species determination. The cytochrome oxidase subunit I (*CO1*) gene has been successfully used in phylogenetic studies on mites [41–44]. However, these genes offer limited information for classifying *Neocypholaelaps* species within the genus and often have limited data in GenBank (https://www.ncbi.nlm.nih.gov) and BOLD systems (https://boldsystems.org). The nuclear ribosomal DNA genes, specifically 18S and 28S, hold promise for resolving higher-level mite phylogenies due to their combination of conserved and variable regions [45–48]. These genes contain conserved regions for identifying deeper branches and variable regions that can be used to discern relationships between closely related taxonomic units [45]. Furthermore, the combination of information from the 18S and 28S genes aids in a more accurate phylogenetic classification than using individual genes alone [47].

In this study, we employed morphological identification methods in conjunction with sequences of the 18S and 28S genes to elucidate the phylogenetic relationships between members of *Neocypholaelaps* sp. KOR23 and their related groups. Additionally, we provided insights into the potential pathogen reservoirs and transmission capabilities of these mites in honeybee colonies. This information offers a comprehensive perspective on the escalating disease prevalence in South Korean honeybees as well as its implications on a global scale.

## Materials and methods

### *Neocypholaelaps* mite collection and morphological identification

*Neocypholaelaps* sp. KOR23 mites were observed on beehive *A. mellifera* colonies in Wonju City, Gangwon Province, ROK, in 2023. Nine colonies from three apiaries were used to collect *Neocypholaelaps* sp. KOR23 mites. These mites were gathered from hive debris and stored in 50 mL tubes (Falcon) containing 95% ethanol that was labeled with hive number. The sample was immediately subjected to total nucleic acid extraction upon arrival the laboratory. The *Neocypholaelaps* mites were identified and imaged using a stereomicroscope (Discovery V8 Stereo, Germany) and a Leica DM1750M microscope (Figs 1 and S1). Identification at the genus level requires that genus classification characteristics are identified according to Delfinado and Baker (1983) [15] and Narita et al. (2013) [30].

### Total nucleic acid extraction and amplification of *CO1*, 18S, and 28S regions

The total nucleic acid (TNA) of the mites were extracted using the Maxwell RSC Viral Total Nucleic Acid Purification Kit (Promega, USA) [14]. The hive debris samples were observed and pooled to collect *Neocypholaelaps* mites under a Discovery V8 Stereo microscope. Three *Neocypholaelaps* mite samples were collected from three apiaries. Ten adult mites from each apiary were transferred onto a Petri dish containing UltraPure™ distilled water (Invitrogen, USA) for washing. Mites were manually collected for extraction of TNA using a mounting needle under a dissecting microscope. Those mites were pooled and placed in 2 mL Eppendorf tubes containing 300 μL phosphate-buffered saline (1x) and 2.381 mm steel beads (Hanam, ROK) and used for TNA extraction [24]. The extracted TNA of *Neocypholaelaps* mite was stored at –20˚C for PCR amplification of *CO1*, 18S, and 28S regions and checking honeybee pathogens.

Primers for the amplification of *CO1*, 18S, and 28S ribosomal DNA genes were designed based on the available sequences in GenBank (https://www.ncbi.nlm.nih.gov) and BOLD

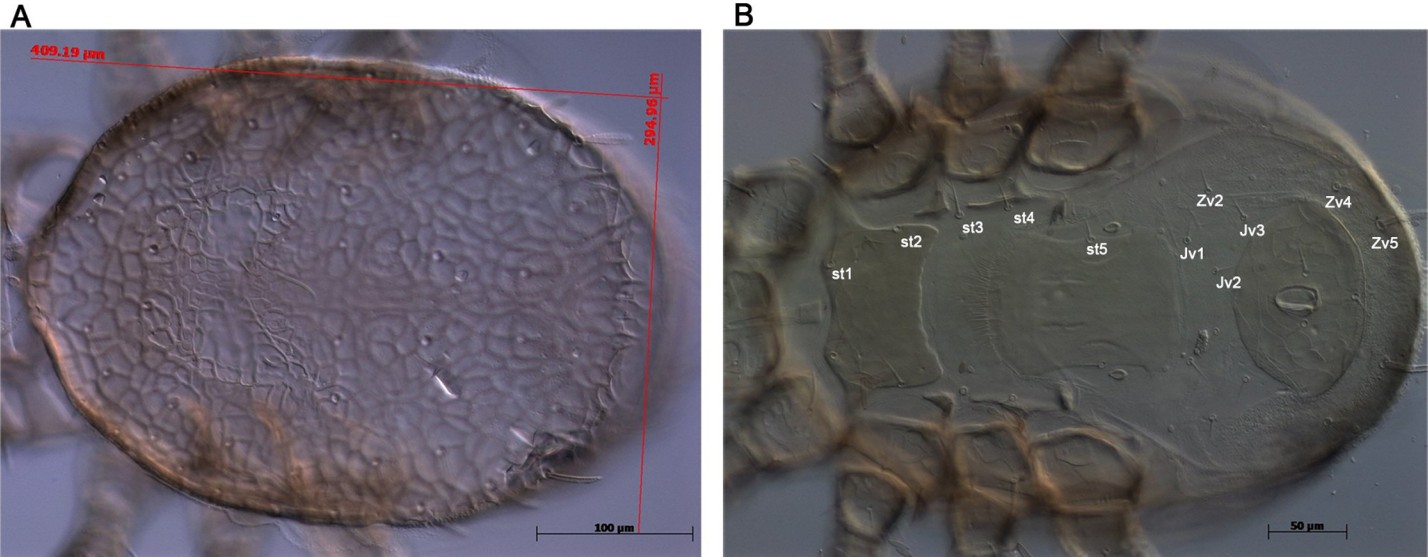

**Fig 1. Morphological characteristics of *Neocypholaelaps* sp. female mite.** Identity of acaroid mites was determined by analyzing morphological characteristics using a phase contrast microscope. (A) Dorsal view of the mite. (B) Ventral view of the mite.

systems (https://boldsystems.org). Based on the morphological characteristics results observed under the microscope, the *Neocypholaelaps* mite was identified in this study. Then, the genetic analysis of *CO1*, 18S, and 28S genes was done for further identification of collected mites. Primers were designed based on the sequence information of *Neocypholaelaps* species available on GenBank and BOLD systems (Table 1). The available primers of *CO1* gene were designed for *N. indica* Evans (1963), *N. apicola* Delfinado-Baker & Baker, 1983, and *Neocypholaelaps* sp. (NCBI accession nos.: LC522089.1, KP966315.1, and MF911280); The 18S and 28S gene were designed for *Neocypholaelaps* sp. (NCBI accession nos.: FJ911807.1 and FJ911742.1). These primers are listed in Table 1. The PCR reaction was performed in 50 μL of AccuPower® PCR preMix and Master Mix (Bioneer, ROK) containing 10 pmol of each primer and 20 ng of DNA as a template. The total volume was increased up to 50 μL with distilled water. The thermocycler protocol by Nguyen et al. was applied [24]. PCR amplification products were visualized by electrophoresis on a 1% agarose gel. The bands were excised under UV light and purified using a QIAquick Gel Extraction Kit (Qiagen, USA). Purified PCR products were sequenced by Cosmogentech (ROK).

## Detection of honeybee pathogens

The total nucleic acid extracted from the mite samples was tested for honeybee pathogens [24, 49]. The honeybee pathogens that were detected in Korean honeybee colonies were employed to assess their presence in *Neocypholaelaps* mite. These pathogens include the following viral pathogens: AFB (American foulbrood), EFB (European foulbrood), ASCO (*Ascosphaera apis*), ASP (*Aspergillus flavus*), *Nosema ceranae*, *Nosema apis*, *Spiroplasma* (*Spiroplasma* sp. and *S. apis*), *Trypanosoma* spp., SBV (sacbrood virus), KSBV (Korea sacbrood virus), DWV (deformed wing virus), BQCV (black queen cell virus), KBV (Kashmir bee virus), ABPV (acute bee paralysis virus), IAPV (Israeli acute paralysis virus), AmFV (*Apis mellifera* filamentous virus), species of Lake Sinai virus (LSV1, LSV2, LSV3, and LSV4), VDV-1 (*Varroa destructor virus* 1), and VDV1-DWV (recombinant *Varroa destructor virus 1* and deformed wing viruses). Honeybee pathogens were detected using RT-qPCR Kits [iNtRON Biotechnology, Inc., ROK], Pobgen bee pathogen detection Kit [Postbio, ROK], SsoAdvanced Universal SYBR Green Supermix [Bio-Rad, USA], and iTaq Universal SYBR Green One-Step Kit [Bio-Rad, USA]. The sequences of primers targeting honeybee pathogens for detection in *Neocypholaelaps* mite samples are provided in S1 Table. Melt-curve dissociation analysis was performed to verify the specificity of the PCR amplification. Negative and positive controls were included for each run. Samples with a $C_t$ value of $\leq 35$ and consistent melting curves were considered positive.

**Table 1. List of primers used for the amplification of *CO1*, 18S, and 28S genes in this study.**

| Strains | Genes | Primers | Sequence (5' → 3') | Size (bp) |
|---|---|---|---|---|
| *Neocypholaelaps* sp. | 18S | Neo18-For | GGATGTGATTAGTTAATTGG | 1590 |
| | | Neo18-Rev | CTTCATTGCAAATAATACAGG | |
| | 28S | Neo28-For | TTAAATACAACAAGAGATGA | 1258 |
| | | Neo28-Rev | CCTGCTGTCTTCAGCACTAAC | |
| *N. indica* | *CO1* | Neo1-For | GGTACTTTATATTTTATTT | 675 |
| | | Neo1-Rev | GGTGACCAAAAAATCAAAAT | |
| *N. apicola* | *CO1* | Neo2-For | GCTCATGCTTTTATTATAAT | 326 |
| | | Neo-Rev | AATAGTACAAATAAAATTAA | |
| *Neocypholaelaps* sp. | *CO1* | Neo3-For | TTATGGTGATGCCTGCTATA | 300 |
| | | Neo-Rev | AATAGTACAAATAAAATTAA | |

## Phylogenetic analysis

The amplified 18S and 28S ribosomal DNA gene sequences of *Neocypholaelaps* sp. KOR23 obtained in this study were deposited in the National Center for Biotechnology Information (NCBI) GenBank database under accession numbers OR576776.1 and OR576780.1, respectively. Those sequences were searched and compared using the nucleotide Basic Local Alignment Search Tool (BLASTn). The identified sequences of *Neocypholaelaps* mite were compared with those of the Mesostigmata mite family deposited in the NCBI database. Nucleotide sequences were aligned using ClustalW alignment in BioEdit version 7.0.0 software [50]. Sequence alignment was performed using the Molecular Evolutionary Genetics Analysis software (MEGA, version 11.0.13), and a phylogenetic tree was constructed based on the neighbor-joining method and a bootstrap probability score of 1000 [51].

## Results

### Morphological identification

The morphological identification of the collected mite showed similar characteristics to the genus of *Neocypholaelaps* adult mite as describe of *N. favus* Ishkawa, 1968, *N. apicola* Delfinado-Baker & Baker, 1983, *N. geonomae* n. sp. Moraes & Narita, 2010, and *N. indica* Evans, 1963 [26–28, 36]. *Neocypholaelaps* adult mite dorsal and ventral surfaces were observed under both microscope discovery V8 Stereo (S1A and S1B Fig) and electron microscopy Leica DM1750M (S1C and S1D Fig) for magnification. The genus *Neocypholaelaps* is characterized by its extremely small size, approximately 2060–2190 μm in length and 1322–1537 μm in width (S1A and S1B Fig). The dorsal shield is entirely reticulate, with reticula formed by simple lines, measuring 409–412 μm in length and 294–307 μm in width (Fig 1A). The sternal shield was wider than it was long, bearing setae *st1* and *st2*, whereas *st3* and *st4* were situated on the unsclerotized cuticle (Fig 1B). The genital shield is smooth, 75–83 μm (average 79 μm) wide at its widest level, slightly convex posteriorly, and bearing *st5*. Moreover, it possesses a pair of pores and a pair of lyrifissures on the unsclerotized cuticle, located posterolateral to *st5*. The anal shield appeared smooth and oval, was equipped with three stouts and barbed setae, and featured a curved transverse line immediately anterior to the cribrum. Opisthogastric setae (*Jv1*–*Jv5* and *Zv2*) were found on the unsclerotized cuticle (Fig 1B). In addition, a sclerotized transverse line was observed immediately posterior to the genital shield margin. Setae *Jv4* and *Jv5* are stout and barbed, whereas other ventral idiosomal setae are setiform and smooth.

### Genetic identification

In this study show that using primers specific for the species *Neocypholaelaps* mites were unable to amplify the *CO1* gene. This indicates that our isolated strains in the ROK do not belong to the known *N. indica* Evans, 1963, *N. apicola* Delfinado-Baker & Baker, 1983, or *Neocypholaelaps* sp. species found in honeybee colonies. Owing to the limited information on *CO1* sequences in GenBank and BOLD systems, it is challenging to distinguish species-level differences based on *CO1* sequences.

We successfully optimized the PCR products for 18S and 28S gene segments of 1590 and 1258 bp, respectively (Fig 2). After removing noisy sequences during read processing, the 18S and 28S sequences were subjected to BLASTn in GenBank, showing the highest similarity to *Neocypholaelaps* sp. APGD-2010 (NCBI accession nos.: FJ911807.1 and FJ911742.1) at 98.81 and 92.17%, respectively. These results demonstrate that the 18S gene region is much more conserved than the 28S region. The sequence similarity of 18S and 28S rRNA indicated a close relationship between this mite and the strain *Neocypholaelaps* sp. APGD-2010.

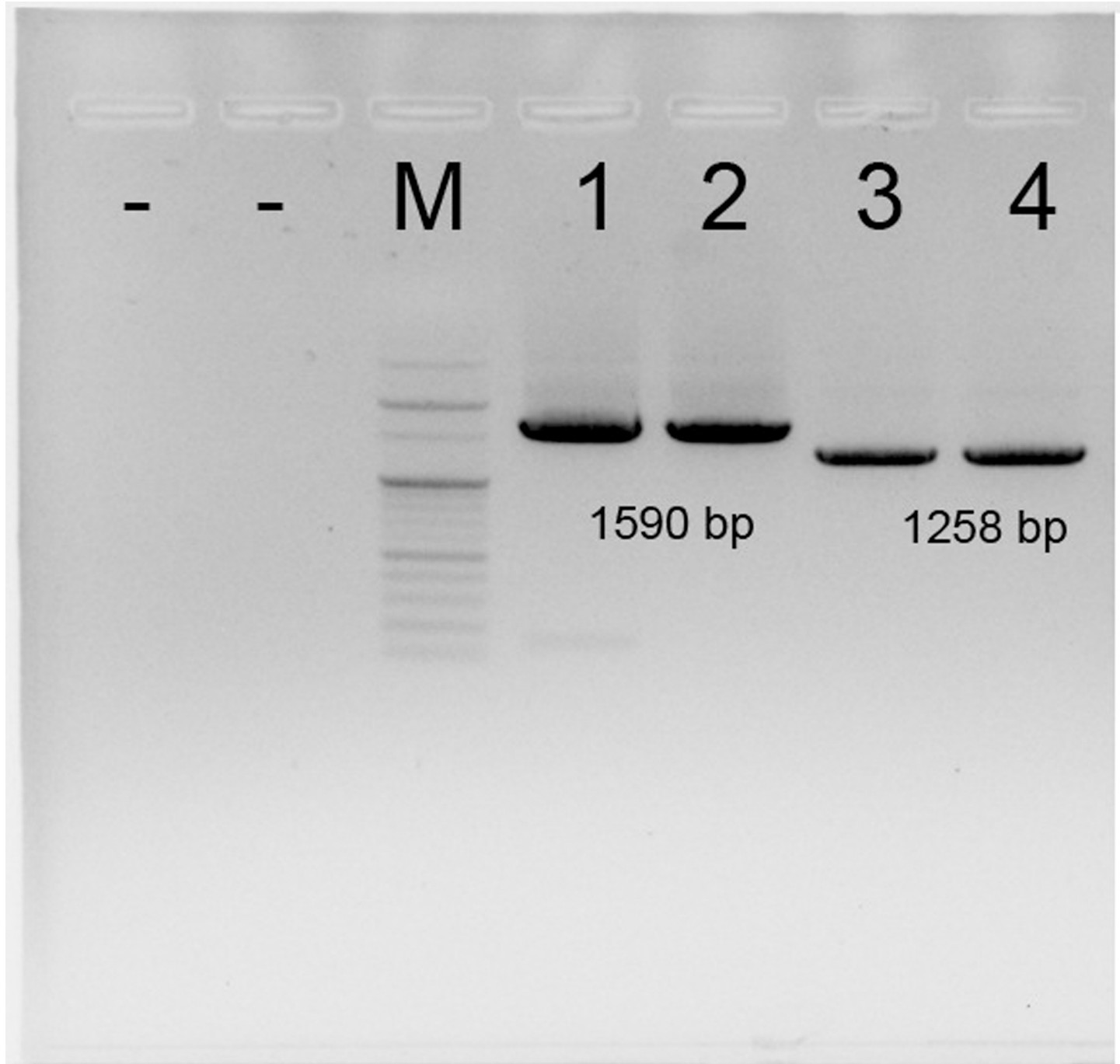

**Fig 2. Amplification of 18S and 28S regions from DNA mite templates.** M is 100 bp DNA ladder (Enzynomics, ROK); 1 and 2 are PCR products of DNA templates using the 18S pair of primer; 3 and 4 are PCR products of DNA templates using the 28S pair of primer; "−" is negative control without a DNA template. Amplicon size from sample DNA is shown in base pair (bp).

## Phylogenetic tree

Based on 18S sequence-based phylogeny, two families (Laelapidae and Ameroseiidae) within the order Mesostigmata were found to be associated with honeybees. The 18S rRNA sequence

of this mite species was similar to that *of Neocypholaelaps* sp. APGD-2010 and *N. ampullula* Berlese, 1910 voucher strain MZLQ6495 (Fig 3). In contrast, the 28S sequence-based phylogeny revealed three families (Laelapidae, Melicharidae, and Ameroseiidae) within the order Mesostigmata that are related to honeybees (Fig 4). Combining the 18S and 28S sequences of this mite species, it was identified as a member of the genus *Neocypholaelaps* but represents a distinct species. Our study newly provides the genetic information of a strain in the genus *Neocypholaelaps* isolated from beehives in the ROK in 2023 and was designed as *Neocypholaelaps* sp. KOR23.

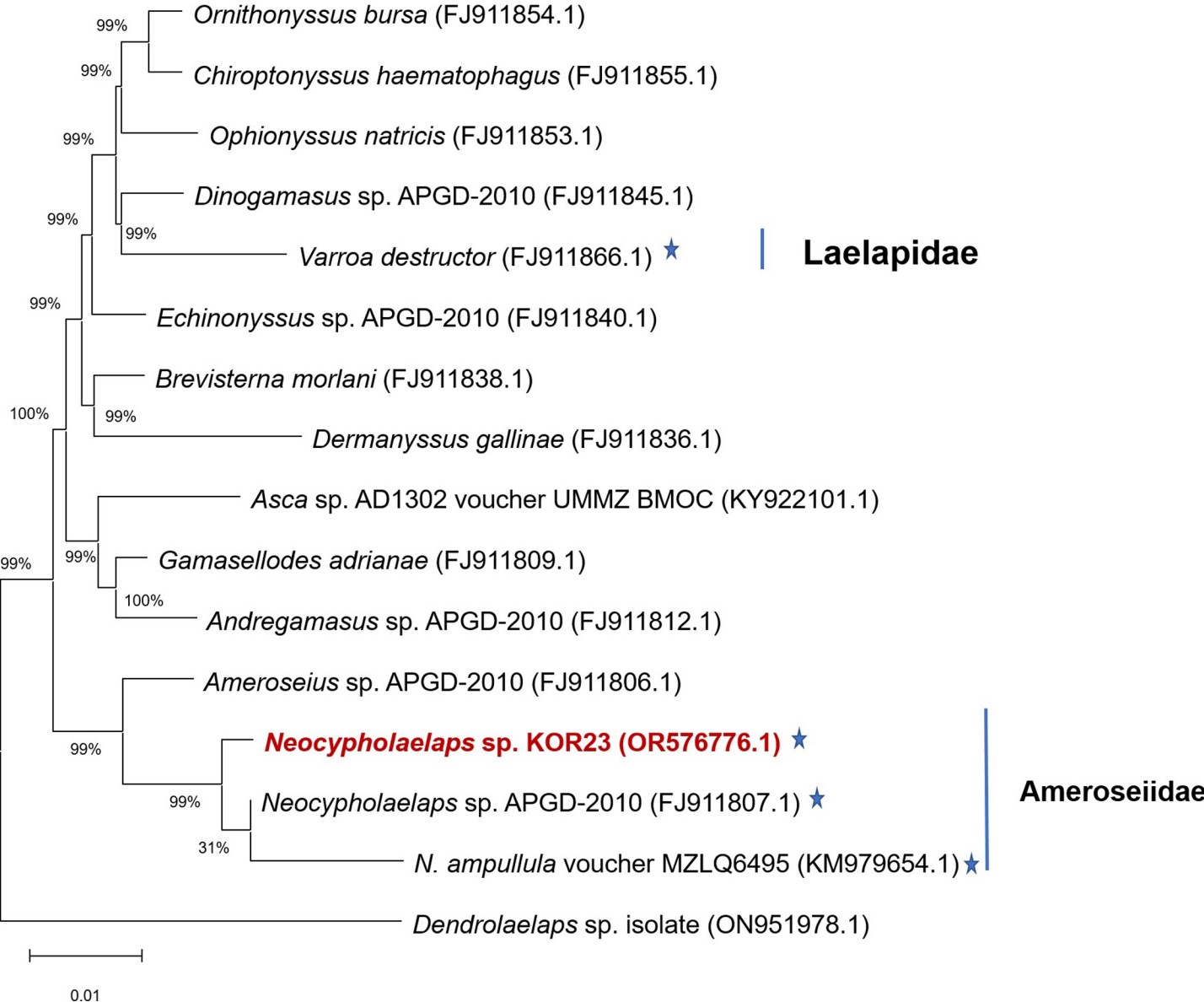

**Fig 3. The phylogenetic tree is based on the 18S sequences.** The used species were selected based on the level of nucleotide sequence similarity within the order Mesostigmata. Star (★) indicates species related to honeybees.

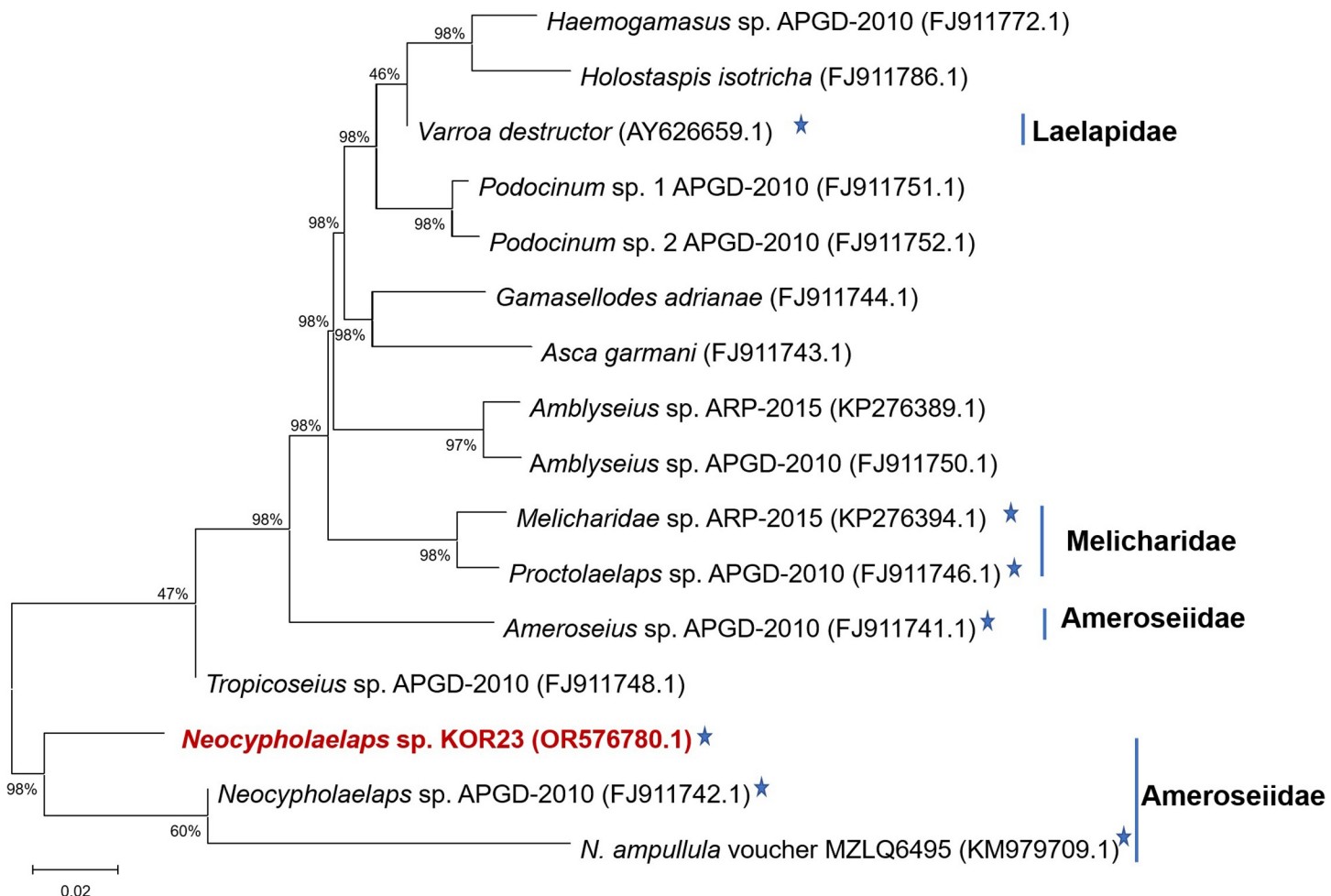

**Fig 4. The phylogenetic tree is based on the 28S sequences.** The used species were selected based on the level of nucleotide sequence similarity within the order Mesostigmata. Star (★) indicates species related to honeybee colonies.

### Testing of honeybee pathogens in *Neocypholaelaps* mites

As certain mite species present in beehives act as parasites on honeybees causing serious impact, this study aimed to investigate the presence of honeybee pathogens within newly isolated *Neocypholaelaps* species in ROK. The identities of the pathogens were confirmed by real-time RT-PCR (Table 2). Nine honeybee pathogens (AFB, ASCO, *Spiroplasma* sp., DWV, BQCV, CBPV, IAPV, KV, and LSV3) were detected in this mite species out of a total of 25 that were screened. Results indicated the presence of bacteria and viruses in *Neocypholaelaps* sp. KOR23 with high *Ct* values. Some honeybee pathogens findings suggest that *Neocypholaelaps* sp. KOR23 has the potential to harbor pathogens and may serve as a vector for honeybee infections, contributing to the rapid increase in disease outbreaks among honeybee colonies in the ROK. However, *Nosema apis* and *N. ceranae* and protozoa *Trypanosoma* spp. have not been identified in *Neocypholaelaps* sp. KOR23.

## Discussion

This study presents in detail the process of identification of *Neocypholaelaps* sp. KOR23 mite within honeybee colonies in Gangwon Province, in ROK. The species-level morphology of

**Table 2. Detection of honeybee pathogens in *Neocypholaelaps* sp. KOR23 mite.**

| Pathogens | Name | Cycle threshold (*) | | | Average |
|---|---|---|---|---|---|
| | | 1 | 2 | 3 | |
| **Bacteria** | AFB | 34.71 | 33.44 | 34.08 | 34.08 ± 0.64 |
| | EFB | - | - | - | - |
| | ASP | - | - | - | - |
| | ASCO | 32.56 | 35.15 | 34.27 | 33.99 ± 1.31 |
| | *Spiroplasma* sp. | 32.25 | 31.12 | 30.95 | 31.44 ± 0.71 |
| | *Spiroplasma apis* | - | - | - | - |
| **Fungi** | *Nosema apis* | - | - | - | - |
| | *Nosema ceranae* | - | - | - | - |
| **Protozoa** | *Trypanosoma* spp. | - | - | - | - |
| **Viruses** | AmFV | - | - | - | - |
| | SBV | 40 | 35.39 | 34.53 | 36.64 ± 2.94 |
| | BQCV | 30.50 | 34.85 | 34.64 | 33.33 ± 2.45 |
| | CBPV | 36.13 | 30.90 | 30.62 | 32.55 ± 3.10 |
| | ABPV | - | - | - | - |
| | IAPV | 34.19 | 33.82 | 23.60 | 30.54 ± 6.01 |
| | KV | 35.05 | 35.27 | 34.02 | 34.78 ± 0.67 |
| | VDV1-DWV | - | - | - | - |
| | VDV-1 | - | - | - | - |
| | DWV | 33.14 | 35.20 | 35.06 | 34.46 ± 1.15 |
| | KSBV | - | - | - | - |
| | KBV | - | - | - | - |
| | LSV1 | - | - | - | - |
| | LSV2 | - | - | - | - |
| | LSV3 | 33.52 | 34.18 | 34.26 | 33.99 ± 0.41 |
| | LSV4 | - | - | - | - |

(*) Data for cycle thresholds ($C_t$) of the three samples. The $C_t$ value of ≤ 35 were considered positive.(-) No detection

*Neocypholaelaps* mites were described and reported [28–30, 38]. According to the morphology of dorsal and ventral of adult mite, the new mite isolated from Korean honeybee colonies was observed belong to genus of *Neocypholaelaps* (Figs 1 and S1). The morphological information of *Neocypholaelaps* is limited in this study, making it challenging to compare with other species. Based on the general morphological and genetic characteristics of 18S and 28S genes, this new mite belongs to the genus *Neocypholaelaps*, with close similarity to *Neocypholaelaps* sp. APGD-2010 that was first isolated from the United States. The discovery of *Neocypholaelaps* in ROK, a temperate region, aligns with recent observations suggesting the genus' ability to adapt to novel climatic conditions and potentially new host associations. The observed variation in the 18S and 28S ribosomal genes suggests they may play a role in adaptation to novel climatic conditions and potentially new host associations. Further research is needed to gain a deeper understanding of the distribution patterns and hosts of *Neocypholaelaps* species. To accurately classify the subspecies level of *Neocypholaelaps* sp. KOR23, detailed morphological characteristics need to be studied, comparing them with strains isolated from temperate regions such as *N. favus* Ishikawa, 1968, and *N. apicola* Delfinado-Baker & Baker, 1983, as reported [34, 38]. Additionally, decoding the entire gene sequence of this mite species would be essential.

In honeybee colonies, bees nesting in cavities play a crucial role as hosts for a diverse range of species, including both pathogenic and non-pathogenic species. The presence of bee mites

can have serious consequences on honeybee populations. Infestation of honeybees with hemo-lymph-feeding mites, such as *Varroa destructor*, *Tropilaelaps*, and *Tyrophagus* mites, has been linked to increased colony loss during the winter season and the transmission of honeybee diseases [6, 14, 23, 24, 52]. The coexistence of these parasitic characteristics within honeybee colonies facilitates the exchange of parasites. Certain mite species in this genus, such as *N. phooni* Baker & Delfinado-Baker, 1985 and *N. malayensis* Delfinado-Baker, Baker & Phoon, 1989 are associated with stingless bees (*Heterotrigona*, *Geniotrigona*, *Tetragonula*, and *Meliponula*). Their life cycle occurs entirely inside the bee colonies, but there is no evidence of parasitism in adult or brood bees [31]. In contrast, in Japan, an astonishing number of 3000 *N. favus* Ishikawa, 1968 mites were found in a single colony, resulting in considerable harm to honeybees [34]. Similarly, as many as 400 *N. indica* Evans, 1972 mites were observed in a single *A. cerana* individual [8]. High infestation rates of *N. indica* Evans, 1972 and its invasion on the body surfaces of three honeybee species were also recorded in India in 2021 [27]. This study identified the presence of *Neocypholaelaps* sp. KOR23 in honeybee colonies within Gangwon province at high densities. Further research is crucial to understand its potential role in honeybee health and develop appropriate management strategies.

Analysis of the presence of honeybee pathogens within these mites showed that nine of twenty-five pathogens were found, indicating the prevalence of honeybee diseases in this mite. The presence of several honeybee pathogens in *Neocypholaelaps* species suggests that these mites pose a risk as potential carriers and transmitters of honeybee diseases. Possibly, this could be one of the factors contributing to severe damage to honeybee colonies. This study is the first to demonstrate the presence of honeybee pathogens in *Neocypholaelaps* sp. mite. Recent research has also reported the appearance of honeybee pathogens in parasites from honeybee colonies. Almost honeybee pathogens have been detected in *Varroa* mite [5, 6, 53, 54]; DWV and BQCV were found on the small hive beetle (*Aethina tunida*) [55]; DWV and *Trypanosoma* spp. were detected on *Tyrophagus curvipenis* [24]. Some of viruses and bacteria pathogen of honeybee were detected in *Neocypholaelaps* sp. KOR23. However, microsporidian (*Nosema apis* and *N. cernaena)* and protozoa (*Trypanosoma* spp.) were not seen in *Neocypholaelaps* sp. KOR23. Further study on the detection of the same pathogens in the honeybee colonies where the mites were detected might be needed to confirm the role of *Neocypholaelaps* sp. KOR23 mite in pathogen transmission.

In conclusion, the novel *Neocypholaelap*s sp. KOR23 mite was detected in honeybee colonies in Gangwon Province, ROK. Our phylogenetic analysis, conducted using the 18S and 28S regions (NCBI accession nos.: OR576776.1 and OR576780.1), enabled the precise identification of the genus of *Neocypholaelaps* mite and its close relationship with *Neocypholaelaps* sp. APGD-2010 (NCBI accession nos.: FJ911807.1 and FJ911742.1). The 18S and 28S genes could be applied for appearance *Neocypholaelaps* sp. from beehive honeybee colonies in different provinces in ROK. The presence of some honeybee pathogens in this mite species highlights the significance of disease transmission within honeybee colonies. However, a thorough analysis and comparison of the presence of the pathogen in this mite and corresponding honeybee colonies is required. Such an analysis will provide a better assessment of infection levels, disease reservoir potential, and the role of the mite as an intermediate vector in the transmission of diseases to honeybee colonies.

## Supporting information

**S1 Fig. Morphological characteristics of *Neocypholaelaps* mite under microscopy.** (A, B) Optical microscope Discovery V8 Stereo (Germany) with a magnification of 5.0× (Dorsal and ventral of *Neocypholaelaps* sp. adult mite). (C, D) Dorsal and ventral view of *Neocypholaelaps*

sp. adult mite under a Leica DM1750M microscope at 10× magnification.
(TIF)

**S1 Table. Primers and probes used for detection of honeybee pathogens.**
(DOCX)

**S1 Raw image.**
(TIF)

## Acknowledgments

We would like to thank Animal and Plant Quarantine Agency for approval to conduct this research. Our heartfelt thanks go to the beekeepers who facilitated the sampling. We extend our appreciation to all members of our laboratories for their unwavering dedication and diligent efforts.

## Author Contributions

**Conceptualization:** Thi-Thu Nguyen, Mi-Sun Yoo, Yun Sang Cho.

**Data curation:** Mi-Sun Yoo, Yun Sang Cho.

**Formal analysis:** Thi-Thu Nguyen, Jong-Ho Lee, A-Tai Truong.

**Methodology:** Thi-Thu Nguyen, Jong-Ho Lee, So-Youn Youn, Se-Ji Lee.

**Writing – original draft:** Thi-Thu Nguyen.

**Writing – review & editing:** A-Tai Truong, Soon-Seek Yoon, Yun Sang Cho.

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
