## [Decision Letter · Decision Letter 0]

4 Dec 2023

PONE-D-23-31153Identification and pathogen detection of a new Neocypholaelaps species (Acari: Mesostigmata: Ameroseiidae) from beehives in the Republic of KoreaPLOS ONE

Dear Dr. Cho,

Thank you for submitting your manuscript to PLOS ONE. After careful consideration, we feel that it has merit but does not fully meet PLOS ONE’s publication criteria as it currently stands. Therefore, we invite you to submit a revised version of the manuscript that addresses the points raised during the review process.

We look forward to receiving your revised manuscript.

Kind regards,

Wolfgang Blenau

Academic Editor

PLOS ONE

Journal Requirements:

"This work was supported by Animal and Plant Quarantine Agency (Grant No. N-1543081-2021-25-03)."

**Additional Editor Comments:**

The Reviewer finds it daring to postulate a new species based on the morphological criteria provided (see below). This point of criticism in particular should be addressed in detail. For example, more morphological details could be provided. Some statements should be supported by references (e.g. pp. 39-41, p. 114ff, etc.). The description of the collection method should be clarified. Relevant sequences should be deposited in GenBank and accession numbers should be provided. Furthermore, some figure legends should be clarified. More details can be found in the Reviewer's report. The Academic Editor would like to apologize once again for the relatively long review period. I explained the reason for this in my last email.

Reviewers' comments:

Reviewer's Responses to Questions

**Comments to the Author**

1. Is the manuscript technically sound, and do the data support the conclusions?

Reviewer #1: Partly

2. Has the statistical analysis been performed appropriately and rigorously? 

Reviewer #1: Yes

3. Have the authors made all data underlying the findings in their manuscript fully available?

Reviewer #1: Yes

4. Is the manuscript presented in an intelligible fashion and written in standard English?

Reviewer #1: Yes

5. Review Comments to the Author

Reviewer #1: In this study, Neocypholaelaps of the genus were found in Western honeybee hives in Korea (ROK). Based on molecular analysis, the authors believe the mite belongs to a new species. Furthermore, the detection of bee pathogens in this mite sample highlights its importance in the spread of bee diseases. Regarding the idea that this species is new, I think the authors should clarify it further and resubmitting it in another article. The following is my opinion:

Title:

Identification and pathogen detection of a Neocypholaelaps species (Acari: Mesostigmata: Ameroseiidae) from beehives in the Republic of Korea

Abstract:

Line 22: Add the scientific name of the host

Line 26-27: It is recommended to delete the description about the new species

Introduction:

Line 39-41: “Although some mite species…..” -> A very ambiguous explanation with no further references to support it.

Line 48: small honeybees -> do you mean dwarf honey bee? Add the scientific name of the bee

Line 49: Add the scientific names of the giant honey bees

M &M

Line 87-88: Add the scientific name of the mite host

Line 88-89: The description of collection method is very vague. The preservation method of pathogen samples is usually not the same as that of identification samples stored in 95% ethanol. This will almost cause the pathogen samples to degrade in a short period.

Line 101-102: Although the sampling method refers to the previous paper, the explanation here is too brief and difficult to understand.

Line 114: The types and primers of pathogen detection should refer to previous papers, but only the types of pathogen detection are mentioned here.

Line 122-123: It is unclear why the author did not detect VDV-1 and LS1 viruses, but to detect VDV-DWV.

Line 133: Relevant sequences should be deposited into the GenBank and given accession numbers.

Line 136-148: It is a bit far-fetched to define a new species in morphology. In the morphological identification, there is no problem with genus-level identification, but there is no description or comparisons at the species level. Even the details in the figures can only be barely identified to the genus. For species-level, the morphological identification did not provide corresponding hand drawings or detailed microscopic photos.

L160 said that based on the morphological characteristics, they designed 18S and 28S primers. The explanation is unclear and I can’t understand the correlation?

Line 175-183: The DNA extraction method and analysis method described in the M&M are rough. It seems that 10 mites are used to form a DNA sample of KOR23. For the genetic analysis, it seems that only two sequences from other Neocypholaelaps species are used. These trees can explain the correct identification of the genus, but the data explain the new species is scarce. Furthermore, the species description of APGD-2010 is also unclear, and it is difficult to explain whether there are any difference in other aspects, such as in morphology.

Line 194 and 198: Two sentences conflict with the results of fungal testing

Discussion

There is a little description and discussion of the pathogen detections. Whether there is relevant pathogen data for the mite species may be used for discuss and comparison.

Figures:

Fig 1: L137 writes the body length and width of Fig1B. The scale should be increased or the length should be marked in this figure. The rest of Fig1 (A,C,D) is unclear, especially A, which feels very redundant.

Figure descriptions should be reconsidered, for example, Figures 1 and 2 do not show explanations of "morphological characteristics”

Fig2 should be added in the caption as female (or maybe another gender or age?)

Further:

For documents on species taxonomy, the scientific names should include the author and the year of publication.

6. PLOS authors have the option to publish the peer review history of their article (what does this mean?). If published, this will include your full peer review and any attached files.

Reviewer #1: **Yes: **I-Hsin Sung

---

## [Author Response · Author response to Decision Letter 0]

28 Jan 2024

Responses to Reviewers’ Comments

We appreciate the reviewers for their invaluable comments. As explained below, we have revised our original manuscript and materials in response to all of the reviewers’ comments. What follows are our point-by-point responses to the comments.

Reviewer #1: In this study, Neocypholaelaps of the genus were found in Western honeybee hives in Korea (ROK). Based on molecular analysis, the authors believe the mite belongs to a new species. Furthermore, the detection of bee pathogens in this mite sample highlights its importance in the spread of bee diseases. Regarding the idea that this species is new, I think the authors should clarify it further and resubmitting it in another article. The following is my opinion:

Title:

Identification and pathogen detection of a Neocypholaelaps species (Acari: Mesostigmata: Ameroseiidae) from beehives in the Republic of Korea

Specific comments:

Abstract:

Line 22: Add the scientific name of the host

Response:

Thank you for your insightful comments. The title was revised according to the suggestion. The information regarding the scientific name of the host (Apis mellifera) was added in the revised manuscript. 

Line 26-27: It is recommended to delete the description about the new species

Response: Modified as suggested.

Introduction:

Line 39-41: “Although some mite species…..” -> A very ambiguous explanation with no further references to support it.

Response: The sentence has been clarified as follow: In A. mellifera, some mite species may not exhibit parasitic behavior (e.g. Tracheal, Forcellinia faini, Melichares dentriticus, Afrocypholaelaps aficana mites) [1–4]. Additionally, specific mite species have been identified as potential vectors of disease to bees (e.g. Varroa and Tropilaelaps mites) [5–7], while a notable number of mite species can induce stress in bees (e.g. Neocypholaelaps indic and Caroglyphus lactis mites) [8,9]. This clarification is supported by relevant references (Lines 38–42)

Line 48: small honeybees -> do you mean dwarf honey bee? Add the scientific name of the bee

Response: The sentences has been revised as suggested: dwarf honeybees (Apis florea)

Line 49: Add the scientific names of the giant honey bees

Response: Sentences were added as suggested: giant honeybees (Apis dorsata)

M &M

Line 87-88: Add the scientific name of the mite host

Response: Modified as suggested. The scientific name of the mite host was honeybee (Apis mellifera)

Line 88-89: The description of collection method is very vague. The preservation method of pathogen samples is usually not the same as that of identification samples stored in 95% ethanol. This will almost cause the pathogen samples to degrade in a short period.

Response: In the methods section at Lines 90 to 94 and Lines 132 to 134, we improved the description of the collection method and elucidated the procedures as follows: 

Neocypholaelaps sp. KOR23 mites were observed on beehive of A. mellifera colonies in Wonju City, Gangwon Province, ROK, in 2023. Nine colonies from three apiaries were used to collect Neocypholaelaps sp. KOR23 mites. These mites were gathered from hive debris and stored in 50 mL Falcon containing 95% ethanol that was labeled with hive number. The sample was immediately subjected to total nucleic acid (TNA) extraction upon arrival the laboratory. (Lines 90–94)

The TNA extracted from the mite samples was tested for honeybee pathogens [25,49]. The honeybee pathogens that were detected in Korean honeybee colonies were employed to assess their presence in Neocypholaelaps mite. These pathogens include the following viral pathogens. (Lines 132–134)

Line 101-102: Although the sampling method refers to the previous paper, the explanation here is too brief and difficult to understand.

Response: the description for nucleic acid extraction was added in the manuscript

“The hive debris samples were observed and pooled to collect Neocypholaelaps mite under microscope Discovery V8 Stereo. Three samples were collected from three apiaries. Ten adult mites from each apiary were transferred onto petri dish containing UltraPure™ distilled water (Invitrogen, USA) for washing. Mites were manually collected for extraction of TNA using a mounting needle under a dissecting microscope. Those mites were pooled and placed in a 2 mL Eppendorf tubes containing 300 μL phosphate-buffered saline (1x) and 2.381 mm steel beads (Hanam, ROK) and used for TNA extraction [25]. The TNA of Neocypholaelaps mite used kept in –20°C for PCR amplification of CO1, 18S, and 28S regions and checking honeybee pathogens.” (Lines 102–110).

Line 114: The types and primers of pathogen detection should refer to previous papers, but only the types of pathogen detection are mentioned here.

Response: Sentences were added more information as suggested: Honeybee pathogens were detected using RT-qPCR Kits (iNtRON Biotechnology, Inc., ROK), Pobgen bee pathogen detection Kit (Postbio, ROK), and iTaq Universal SYBR Green One-Step Kit (Bio-Rad, USA). The primers of pathogens detection were used following by Truong et al. and Nguyen et al. [25,49]. (Lines 142–145)

Line 122-123: It is unclear why the author did not detect VDV-1 and LS1 viruses, but to detect VDV-DWV.

Response: The Varroa destructor virus-1 (VDV-1) and Lake Sinai Virus 1 (LSV1) pathogens were tested and added more information in Table 2.

Line 133: Relevant sequences should be deposited into the GenBank and given accession numbers.

Response: Thank you for your suggestion. The 18S and 28S genes of Neocypholaelaps sp. KOR23 were deposited in an NCBI database with accession numbers: OR576776.1 and OR576780.1 (Lines 149–151)

Line 136-148: It is a bit far-fetched to define a new species in morphology. In the morphological identification, there is no problem with genus-level identification, but there is no description or comparisons at the species level. Even the details in the figures can only be barely identified to the genus. For species-level, the morphological identification did not provide corresponding hand drawings or detailed microscopic photos.

Response: The sentence mentioning the genus level of morphological identification was added in Lines 161 to 168. For species identification the molecular method relying on CO1, 18S, and 28S DNA was used. 

L160 said that based on the morphological characteristics, they designed 18S and 28S primers. The explanation is unclear and I can’t understand the correlation?

Response: Revision was done for the sentence to clarify the intended meaning in the material and methods section from Lines 113 to 121

“Based on the morphological characteristics results observed under the microscope, the Neocypholaelaps mite was identified in this study. Then, the genetic analysis of CO1, 18S, and 28S genes was done for further identification of collected mites. Primers were designed based on the sequence information of Neocypholaelaps species available on GenBank and BOLD systems (Table 1). The available primers of CO1 gene were designed for N. indica Evans (1963), N. apicola Delfinado-Baker & Baker, 1983, and Neocypholaelaps sp. (NCBI accession nos.: LC522089.1, KP966315.1, and MF911280); The 18S and 28S gene were designed for Neocypholaelaps sp. (NCBI accession nos.: FJ911807.1 and FJ911742.1).”

Line 175-183: The DNA extraction method and analysis method described in the M&M are rough. It seems that 10 mites are used to form a DNA sample of KOR23. For the genetic analysis, it seems that only two sequences from other Neocypholaelaps species are used. These trees can explain the correct identification of the genus, but the data explain the new species is scarce. Furthermore, the species description of APGD-2010 is also unclear, and it is difficult to explain whether there are any difference in other aspects, such as in morphology.

Response: Thank you for your comments, we revised the last sentence of the paragraph to be more appropriate with the result identified in this study: “Our study newly provides the genetic information of a strain in the genus Neocypholaelaps isolated from beehives in the ROK in 2023 and was designated as Neocypholaelaps sp. KOR23”. In addition, the term “new species” is removed from the whole manuscript.

Line 194 and 198: Two sentences conflict with the results of fungal testing

Response:

 Thank you for your suggestion. Those sentences were corrected at Lines 220 to 228. “The identities of the pathogens were confirmed by real-time RT-PCR (Table 2). Nine honeybee pathogens (AFB, ASCO, Spiroplasma sp., DWV, BQCV, CBPV, IAPV, KV, and LSV3) were detected in this mite species out of a total of 25 that were screened. Results indicated the presence of bacteria and viruses in Neocypholaelaps sp. KOR23 with high Ct values. Some honeybee pathogens findings suggest that Neocypholaelaps sp. KOR23 has the potential to harbor pathogens and may serve as a vector for honeybee infections, contributing to the rapid increase in disease outbreaks among honeybee colonies in the ROK. However, Nosema apis, N. ceranae, and Trypanosoma spp. have not been identified in Neocypholaelaps sp. KOR23”.

Discussion

There is a little description and discussion of the pathogen detections. Whether there is relevant pathogen data for the mite species may be used for discuss and comparison.

Response: Sentences were added for discussion about the pathogen detection in this section from Lines 266 to 277 “Possibly, this could be one of the factors contributing to severe damage to honeybee colonies. This study is the first to demonstrate the presence of honeybee pathogens in Neocypholaelaps sp. mite. Recent research has also reported the appearance of honeybee pathogens in parasites from honeybee colonies. Various honeybee pathogens have been detected in Varroa mite [5,6,54,55]; DWV and BQCV were found on the small hive beetle (Aethina tumida) [56]; DWV and Trypanosoma spp. were detected on Tyrophagus curvipenis [25]. Some viral and bacterial pathogens of honeybees were detected in Neocypholaelaps sp. KOR23. However, microsporidian (Nosema apis and N. cernaena) and protozoa (Trypanosoma spp.) were not seen in Neocypholaelaps sp. KOR23. Further study on the detection of the same pathogens in the honeybee colonies where the mites were detected might be needed to confirm the role of Neocypholaelaps sp. KOR23 mite in pathogen transmission.”

Figures:

Fig 1: L137 writes the body length and width of Fig1B. The scale should be increased or the length should be marked in this figure. The rest of Fig1 (A, C, D) is unclear, especially A, which feels very redundant.

Figure descriptions should be reconsidered, for example, Figures 1 and 2 do not show explanations of "morphological characteristics”

Fig2 should be added in the caption as female (or maybe another gender or age?)

Response: Thank you for your comment and suggestion about the morphology figure. We moved the Fig 1 to Supplementary data as S1 Fig, and Fig 2 was changed to be Fig 1 in the revised manuscript. The adult female was added in the caption as commented

Fig 1. Morphological characteristics of Neocypholaelaps sp. adult female mite. Identity of acaroid mites was determined by analyzing morphological characteristics using a phase contrast microscope. (A) Dorsal view of the mite. (B) Ventral view of the mite.

---

## [Decision Letter · Decision Letter 1]

13 Feb 2024

PONE-D-23-31153R1Identification and pathogen detection of a Neocypholaelaps species (Acari: Mesostigmata: Ameroseiidae) from beehives in the Republic of KoreaPLOS ONE

Dear Dr. Cho,

Thank you for submitting your manuscript to PLOS ONE. After careful consideration, we feel that it has merit but does not fully meet PLOS ONE’s publication criteria as it currently stands. Therefore, we invite you to submit a revised version of the manuscript that addresses the few minor points raised during the review process. Once this is done, I can accept the manuscript without involving reviewers again.

We look forward to receiving your revised manuscript.

Kind regards,

Wolfgang Blenau

Academic Editor

PLOS ONE

Journal Requirements:

Reviewers' comments:

Reviewer's Responses to Questions

**Comments to the Author**

1. If the authors have adequately addressed your comments raised in a previous round of review and you feel that this manuscript is now acceptable for publication, you may indicate that here to bypass the “Comments to the Author” section, enter your conflict of interest statement in the “Confidential to Editor” section, and submit your "Accept" recommendation.

Reviewer #1: All comments have been addressed

2. Is the manuscript technically sound, and do the data support the conclusions?

Reviewer #1: Yes

3. Has the statistical analysis been performed appropriately and rigorously? 

Reviewer #1: Yes

4. Have the authors made all data underlying the findings in their manuscript fully available?

Reviewer #1: Yes

5. Is the manuscript presented in an intelligible fashion and written in standard English?

Reviewer #1: Yes

6. Review Comments to the Author

Reviewer #1: There are minor issues should be check. Please refer and consider making appropriate revisions. I think it can be accepted in the future.

L40: delete the word “Tracheal,”

L41, L43: possibility of spelling errors on scientific names “Afrocypholaelaps aficana”, “Neocypholaelaps indic” and “Caroglyphus lactis”, please check.

L45-53: In this paragraph, four genera of mites are described as being particularly harmful to bees, but a very abrupt description of Tyrophagus is added in L53-54.

L55-56: states that Neocypholaelaps are mainly distributed in tropical regions, but most of the species later exemplified in paragraphs L56-69 are distributed in subtropical and temperate regions. Further possible reason or additional explanations should be added.

L75&77: The terminology of L75&77 "higher level classification" may be worth discussing. The author may have intended to express a classification closer to the genus or species level, but the term may also be easily confused with the advanced classification methods used to describe family or above levels.

L93 “50 mL Falcon”-> “50 mL tubes (Falcon)”

L244 "with several reported subspecies" It would be better if there are relevant literature guidance

L261-263: It can be described that KOR23 is densely populated and may cause harm to bee colonies, but rather than saying "leading to damage", perhaps it is more conservative to say "leading to threaten" or "leading to weaken"? It can also be slightly blended with the conclusion of the last paragraph of the discussion.

7. PLOS authors have the option to publish the peer review history of their article (what does this mean?). If published, this will include your full peer review and any attached files.

Reviewer #1: No

---

## [Author Response · Author response to Decision Letter 1]

20 Feb 2024

Responses to Reviewers’ Comments

We appreciate the reviewers for their invaluable comments. As explained below, we have revised our original manuscript and materials in response to all of the reviewers’ comments. What follows are our point-by-point responses to the comments.

Reviewer #1: There are minor issues should be check. Please refer and consider making appropriate revisions. I think it can be accepted in the future.

L40: delete the word “Tracheal,”

Response: I deleted it.

L41, L43: possibility of spelling errors on scientific names “Afrocypholaelaps aficana”, “Neocypholaelaps indic” and “Caroglyphus lactis”, please check.

Response: These have been corrected as follows:

 “Afrocypholaelaps aficana” � “Afrocypholaelaps africana”

 “Neocypholaelaps indic” � “Neocypholaelaps indica”

 “Caroglyphus lactis” � “Carpoglyphus lactis”

L45-53: In this paragraph, four genera of mites are described as being particularly harmful to bees, but a very abrupt description of Tyrophagus is added in L53-54.

Response: I thank the reviewer for the reasonable comment. I corrected these sentences more logically and consistently in lines 46-56.

“Obligate honeybee parasites like Varroa (Varroidae, Mesostigmata), Euvarroa (Varroidae, Mesostigmata), Tropilaelaps (Laelapidae, Mesostigmata), and Acarapis (Tarsonemidae, Prostigmata inflict direct harm and facilitate disease spread [10–12]. These mites have co-evolved with their hosts, developing an intimate dependence on honeybees for survival. While Varroa mites primarily parasitize bees in their brood cells [6,13], Euvarroa and Tropilaelaps demonstrate host specificity towards Dwarf honeybees (Apis florea) and giant honeybees (Apis dorsata), respectively [14–18]. The most pathogenic species within the genus Acarapis primarily parasitize A. mellifera but have also been found in Asian honeybee species [19–23]. Interestingly, the genus Tyrophagus (Acaridae: Sarcoptiformes) adds to the list of potentially detrimental mites, displaying parasitic behavior detrimental to both honeybees and bumblebees [24,25].”

L55-56: states that Neocypholaelaps are mainly distributed in tropical regions, but most of the species later exemplified in paragraphs L56-69 are distributed in subtropical and temperate regions. Further possible reason or additional explanations should be added.

Response: We have included this information in the introduction (lines 57-68) and the discussion section (lines 253-259)

L75&77: The terminology of L75&77 "higher level classification" may be worth discussing. The author may have intended to express a classification closer to the genus or species level, but the term may also be easily confused with the advanced classification methods used to describe family or above levels.

Response: This has been corrected (Line 79).

“However, these genes offer limited information for classifying Neocypholaelaps species within the genus”

L93 “50 mL Falcon”-> “50 mL tubes (Falcon)”

Response: I corrected it.

L244 "with several reported subspecies" It would be better if there are relevant literature guidance

Response: I have included this information in lines 259-263.

L261-263: It can be described that KOR23 is densely populated and may cause harm to bee colonies, but rather than saying "leading to damage", perhaps it is more conservative to say "leading to threaten" or "leading to weaken"? It can also be slightly blended with the conclusion of the last paragraph of the discussion.

Response: This expression has been corrected with further research needed to confirm the damage of Neocypholaelaps mite infestation (Lines 279-281).

---

## [Editor Report · Decision Letter 2]

21 Feb 2024

Identification and pathogen detection of a Neocypholaelaps species (Acari: Mesostigmata: Ameroseiidae) from beehives in the Republic of Korea

PONE-D-23-31153R2

Dear Dr. Cho,

We’re pleased to inform you that your manuscript has been judged scientifically suitable for publication and will be formally accepted for publication once it meets all outstanding technical requirements.

Kind regards,

Wolfgang Blenau

Academic Editor

PLOS ONE
---

## [Editor Report · Acceptance letter]

1 Apr 2024

PONE-D-23-31153R2 

PLOS ONE

Dear Dr. Cho, 

I'm pleased to inform you that your manuscript has been deemed suitable for publication in PLOS ONE. Congratulations! Your manuscript is now being handed over to our production team.

Kind regards, 

on behalf of

Dr. Wolfgang Blenau 

Academic Editor

PLOS ONE